# Trauma of Peripheral Innervation Impairs Content of Epidermal Langerhans Cells

**DOI:** 10.3390/diagnostics12030567

**Published:** 2022-02-23

**Authors:** Giovanna Albertin, Barbara Ravara, Helmut Kern, Sandra Zampieri, Stefan Loefler, Christian Hofer, Diego Guidolin, Francesco Messina, Raffaele De Caro, Mauro Alaibac, Ugo Carraro

**Affiliations:** 1Interdepartmental Research Center of Myology (CIR-Myo), University of Padova, 35131 Padova, Italy; giovanna.albertin@unipd.it (G.A.); barbara.ravara@unipd.it (B.R.); sanzamp@unipd.it (S.Z.); 2Department of Neuroscience (DNS), Section of Human Anatomy, University of Padova, 35131 Padova, Italy; diego.guidolin@unipd.it (D.G.); raffaele.decaro@unipd.it (R.D.C.); 3Department of Biomedical Sciences (DSB), University of Padova, 35131 Padova, Italy; ugo.carraro@unipd.it; 4A&C M-C Foundation for Translational Myology, 35100 Padova, Italy; 5Department of Surgery, Oncology and Gastroenterology (DISCOG), University of Padova, 35131 Padova, Italy; 6Ludwig Boltzmann Institute for Rehabilitation Research, Neugebäudeplatz 1, 3100 St. Pölten, Austria; helmut@kern-reha.at (H.K.); stefan.loefler@rehabilitation.lbg.ac.at (S.L.); christian.hofer@rehabilitation.lbg.ac.at (C.H.); 7Institute of Physical Medicine and Rehabilitation, 3300 Amstetten, Austria; 8Department of Medicine (DIMED), Section of Dermatology Clinic, University of Padova, 35131 Padova, Italy; francescomexina@gmail.com

**Keywords:** Langerhans cells, epidermis atrophy, skin biopsy, spinal cord injury, lower or upper motor neuron injury, electrical stimulation

## Abstract

Langerhans cells represent the first immune cells that sense the entry of external molecules and microorganisms at the epithelial level in the skin. In this pilot case-study, we evaluated Langerhans cells density and progression of epidermal atrophy in permanent spinal cord injury (SCI) patients suffering with either lower motor neuron lesions (LMNSCI) or upper motor neuron lesions (UMNSCI), both submitted to surface electrical stimulation. Skin biopsies harvested from both legs were analyzed before and after 2 years of home-based Functional Electrical Stimulation for denervated degenerating muscles (DDM) delivered at home (h-bFES) by large anatomically shaped surface electrodes placed on the skin of the anterior thigh in the cases of LMNSCI patients or by neuromuscular electrical stimulation (NMES) for innervated muscles in the cases of UMNSCI persons. Using quantitative histology, we analyzed epidermal thickness and flattening and content of Langerhans cells. Linear regression analyses show that epidermal atrophy worsens with increasing years of LMNSCI and that 2 years of skin electrostimulation reverses skin changes, producing a significant recovery of epidermis thickness, but not changes in Langerhans cells density. In UMNSCI, we did not observe any statistically significant changes of the epidermis and of its content of Langerhans cells, but while the epidermal thickness is similar to that of first year-LMNSCI, the content of Langerhans cells is almost twice, suggesting that the LMNSCI induces an early decrease of immunoprotection that lasts at least 10 years. All together, these are original clinically relevant results suggesting a possible immuno-repression in epidermis of the permanently denervated patients.

## 1. Introduction

The epidermis of the skin is a cornfield, stratified, and squamous epithelium that forms a barrier isolating the internal fluids of the body and protecting it against various external negative influences. The epidermal compartment contains different appendages that are essential for protection, camouflage, and thermoregulation. The adult epidermis undergoes constant replacement and repair by a resident population of stem cells capable of proliferating and responsible for the homeostasis of the various epidermal compartments. Keratinocytes are the predominant cells of epidermis, which originate in the basal layer and are in constant transition from the *stratum basale* to the *stratum corneum*.

The innate and adaptive immune responses of the skin are deeply intertwined and several cell types are functionally bound to cooperate in order to achieve a protective immune response [1,2]. This complex network composed of cells and molecules has been defined as “skin immune system”. Its cellular components are: keratinocytes, dendritic cells, macrophages, mast cells, granulocytes, and lymphocytes. We focused our analyses on dendritic cells, in particular on Langerhans cells and the relationship with long-term complete lower motor neuron traumatic lesion (permanent denervation). Dendritic cells are myeloid cells specialized in the induction of immune responses, being endowed with the ability to capture foreign antigens and present them to lymphocytes, as well as produce cytokines that shape the inflammatory milieu. In the epidermis, a particular subset, the Langerhans cells, represents the first immune cells that sense the entry of external molecules and microorganisms at the epithelial level [3].

We previously showed that in patients with permanent lower motor neuron lesions (LMNSCI), the severely atrophic quadriceps muscles were recovered by functional electrical stimulation performed at home (h-bFES) [4,5]. Notably, a marked amelioration has been observed after 2 years of h-bFES performed using skin electrodes and an electrical stimulator designed [6,7,8,9,10,11,12,13,14,15]. In particular, we demonstrated that SCI induced worsening of epidermal trophism from 1 to 8 years after lesion and that those changes were reversed after 2 years of electrical stimulation, [13,14,15,16] as described in other skin disorders [17,18,19,20,21].

In this pilot case-study, we evaluated Langerhans cells density and progression of epidermal atrophy in permanent spinal cord injury (SCI) patients suffering with either lower motor neuron lesions (LMNSCI) or upper motor neuron lesions (UMNSCI), both submitted to surface electrical stimulation.

## 2. Materials and Methods

The results of this study concern two groups of SCI patients. The first group included 13 LMNSCI patients of the EU Project RISE. The characteristics of the patients are described in our previous publications [8,9,22].

The second group was composed of three UMNSCI patients; those characteristics are summarized in Table 1.

We certify that all applicable rules concerning the ethical use of human volunteers were followed during the research (approval of ethical committee, Vienna, Austria: Ethikkommission der Stadt Wien Austria: EK-02-068-0702) and that all patients signed an informed consent.

Clinical and functional assessments, follow-up, and tissue biopsies were conducted at the Wilhelminenspital, Vienna, Austria. Electrical stimulation, needle electromyography, brain motor control assessment, and both transcranial and lumbosacral magnetic stimulation were performed to evaluate the complete denervation of quadriceps muscles of both legs in LMNSCI patients, both before and after 2 years of h-bFES [21,22]. These patients were instructed to perform at home the Vienna FES protocol for denervated degenerating muscles (DDM) (h-bFES, see below).

The second group of three UMNSCI was instructed to perform at home electrical stimulation of their innervated leg muscles by surface Neuro Muscular Electrical Stimulation (NMES) (1.0 ms at up to ±80 V and up to ±250 mA.) Mean stimulation current of 128  ±  16 mA and voltage of 39  ±  14 V, as described in references [23,24,25,26,27,28,29,30].

The functional adjusted protocol of LMNSCI, which is crucial to reach the improvement in performances and dimensions of the thigh muscle and in the skin structure, are fully described in a previous article [7].

NMES in the UMNSCI patients was performed with the same electrodes as above described, but the current applied was that typical for innervated muscles, that is 0.6 ms up to ±250 mA and up to ±80 V, but the mean stimulation current was 128  ±  16 mA and voltage of 39  ±  14 V. Each repetition (i.e., series of NMES induced tetanic muscle contractions) was evoked by a 3.5 s train (60 Hz) of electrical pulses (rectangular, biphasic, width 0.6 ms). Consecutive contractions of the same thigh were separated by 4.5 s intervals [9,10,11].

At enrollment and again after about 2 years of h-bFES or NMES, 64 skin biopsies (6 mm diameter) were performed to allow safe collection of muscle tissues in an area where electrodes were routinely applied. The locations of the skin biopsies were the same in all patients, but the post Electrical Stimulation (ES) biopsy was located 3 mm medial from the scar of the first biopsy. Their harvesting at the post ES stimulation protocol spanned from 15 months to 32 months (mean duration 26 ± 1.50 SEM). The skin biopsies were harvested without restrictions, following prudent clinical criteria [7].

The skin biopsies were fixed in 10% buffered formalin and embedded in paraffin. Histological sections of 5 µm thickness were then collected and stained by a standard Hematoxylin-Eosin (H-E) stain protocol and with a standard immunohistochemical procedure to analyze CD1a, a specific marker of Langerhans cells (LCs).

CD1a-positive LCs detection was performed using Dako Autostainer/Autostainer Plus (Dako, Milan, Italy) and CD1a antibodies (M3571 anti-human CD1a, Dako) diluted 1:100 in EnVision buffer using a standard approach [16,30].

The epidermis thickness, interdigitation index, CD1a immunoreactivity (IR%), and density (n/mm^2^) were evaluated by computer-assisted imaging following the procedure previously described (16). Next is Figure 1.

Statistical analyses were performed using GraphPad Prism 5.0 software (GraphPad software, La Jolla, CA, USA). For each leg, representative values of the parameters before and after NMES treatment were obtained by averaging the values taken from the corresponding biopsies and a paired Student’s *t* test was applied to compare the two sets of measurements leg-by-leg. Unpaired *t* test compares UMNSCI date with LMNSCI and on sub-groups of data, obtained by grouping samples according LMNSCI time intervals (<1 year and 5.4–7.6 years). Statistical significance was always *p* < 0.05. We used SPSS rel.13.0, SPSS Inc., Chicago, IL, USA to perform multiple regression analysis of each estimated parameter (epidermis thickness, interdigitation index, Langerhans IR% and density) and two predictor variables (‘age’ and ‘injury time’) on LMNSCI patients before the h-bFES. From the obtained correlation matrix, dominance analysis was also performed by using a specific SPSS plugin [31]. Dominance analysis is a method aimed at evaluating the relative importance of each predictor in a multiple regression [32], and Johnson’s weights were the statistical parameters used to quantify the relative importance of the two predictors here considered [33].

## 3. Results

In a previous paper we definitively proved that 2 years of h-bFES reversed the process of epidermis atrophy and flattening that occurs in between 1 and 8 years post-SCI [6]. Here we show that the age of the LMNSCI patients at enrollment (i.e., at pre h-bFES) might be an important factor influencing treatment outcome. Indeed, the multiple regression analyses (Table 2 and Figure 2) indicated that the baseline (i.e., at the enrollment) epidermal changes exhibited a moderate (R coefficient lower than 0.7) but significant negative correlation with both time from injury and age, suggesting a worsening of the process of epidermis atrophy with time.

We also investigated the relative weight of permanent denervation vs life style imposed to patient by SCI; specifically, if the observed epidermis changes were related to long-term denervation of tissues or to the long-term daily use of wheelchairs. Because no changes could be imposed to the life-style of patients, we compared epidermis characteristics of the two groups of patients, taking advantage of the collection of skins from UMNSCI patients, who have a similar life style, but whose paralyzed legs remain innervated by the lower motor neurons.

Two UMNSCI patients (UMNSCI.1 and UMNSCI.2) analyzed 2–5 years from injury showed a slight increase of epidermis thickness after 2 years of NMES, while the third patient (UMNSCI.3) had possibly a lower response when NMES was started 20 years after UMNSCI. Anyhow, there were no significant changes in epidermis thickness when all data were averaged (Table 3, and Figure 1 an example of H-E staining).

Table 4 shows that averaging all the data of the UMNSCI patients (Pre + Post NMES) and comparing the mean with the mean of all the pre h-bFES of LMNSCI (16), the latter was 17% thinner.

When the mean of the UMNSCI was compared with the subgroup of older LMNSCI patients (within 5–7 years of SCI), the difference increased to 46%, while it disappeared almost completely when the comparison was made with the subgroup of LMNSCI within 1 year of SCI. All together this suggests that thickness of the epidermis of LMNSCI during the first year of injury is almost normal despite denervation, as it appears in the cases of the UMNSCI patients, at least up to 20 years of SCI.

These behaviors are also shown in Table 5 and Table 6, where the results of the analyses of the Interdigitation Index are presented for each patient. There were no significant changes in the Interdigitation index when all data were averaged (Table 5). When the average of all the data of the UMNSCI patients (Pre + Post NMES) was compared with the mean of all the pre h-bFES of LMNSCI (16), the interdigitation index was 20% less (*p* < 0.0001, Table 6).

Analysis of Langerhans cells after NMES by CD1a immunostaining on UMNSCI shows that there are no changes on IR% and LC density and only a minor statistically significant increase on the LC density of them (Figure 3). An example of CD1a immunostaining was shown in Figure 1.

On the other hands, Figure 4 shows that these parameters are double the values present even in very early (less than 1 year) LMNSCI patients.

If we used the average value of UMNSCI pre + post, the analysis shows that the IR% and density are significantly lower on LMNSCI and in particular double IR% on UMNSCI than the value observed in very early (less than 1 year) LMNSCI patients (Figure 4).

## 4. Discussion

One of the main findings of the present study was that skin folds in UMNSCI patients were similar to those observed in normal young adults after 5–20 years of UMN. Moreover, the epidermis of UMNSCI was thicker than that of patients with LMNSCI with the exception of LMNSCI patients after less than 1 year of their injury. The results on the epidermis flattening are particularly important because we know from the literature the index values of young normal persons [34], and thus, we may conclude that both early LMNSCI and even long-term UMNSCI do not implicate epidermis flattening, which is a major mechanism of potential damage in long-term LMNSCI.

Another interesting finding of our study was that the content of LCs in patients from 5 to 20 years after UMNSCI was double that detected in patients with LMNSCI even after less than 1 year from the initial lesion. Therefore, we can conclude that “peripheral motor denervation in LMNSCI” is accompanied by an early “non-progressive” decrease in Langerhans cells content, a behavior that is found for many organ functions in SCI (including muscle atrophy). Indeed, it is well established that trauma to the spine causes functional changes that potentially accelerate the decline of the body functions soon after lesion, then, aging appears to proceed normally [35,36].

Our current analyses show no statistically significant changes in epidermal thickness and Langerhans cell content in three UMNSCI patients, i.e., with peripheral innervation of the thighs. Furthermore, the epidermal thickness and complexity of UMNSCI patients are similar to that observed in LMNSCI during the first year after injury and in the range of normal values [37], while Langerhans cell content is nearly double.

We suggest that permanent denervation induces an early decrease in epidermal immune protection that lasts at least 10 years after LMNSCI, the longest period we could study. However, in full LMNSCI, the well-documented 2-year beneficial effects of surface h-bFES on muscle fiber strength, thigh muscle volume, and size are extended to the epidermis, suggesting that electrical stimulation with anatomically shaped electrodes attached to the skin is also clinically relevant to counteract epidermal atrophy, but not the decrease in number and size of Langerhans cells that remains at the low levels observed in LMNSCI patient after less than 1 year from the initial lesion.

To avoid ethical difficulties of the clinical human trials, studies are also implemented in large mammals, usually pigs, in experimental models of SCI [38]. In a series of studies on the pathogenesis and managements of skin pressure ulcers, Intermittent Electrical Stimulation was demonstrated effectively in the prevention and managements of pressure ulcers [39,40,41,42]. Unfortunately, the main focus was always on deep tissue injuries and much less on the skin. The clinical implications of the alteration in the content of Langerhans cells are unclear. Langerhans cells are able to present antigens locally to activated CD4 T cells that infiltrate the site and through this mechanism are responsible for the induction of allergic contact dermatitis [43]. It is plausible that activation of CD4 T cells may be impaired as a result of denervation with the consequent abolition of the contact sensitivity response. Similarly, in atopic dermatitis, LCs depletion can help to relieve the disease. On the other hand, it has been observed in melanoma that the decrease in the number of LCs may promote the progression of this neoplastic disorder [44]. Initiating tumor immunity may, therefore, be critically dependent on the proper activity of LCs as antigen presenting cells through the ability to induce T cell activation and proliferation. To this regard, a recent study demonstrated a lesser number of LCs in squamous cell carcinoma if compared to basal cell carcinoma, which is a less aggressive subtype of non-melanoma skin cancer [45]. With regard to psoriasis, the role and function of LCs is controversial, as in some subtypes of psoriasis, reduction of LCs is associated with worsening of the disease, whereas in others an improvement of this condition is observed [46]. The role of LCs in response to infections is varied [47]. In particular, there is a variability in LCs response to viral infection, which is determined by the epidermal cell type infected, differences in viral life cycle, and gene expression in those cells. To this regard, a general protective effect is observed in epitheliotropic viruses, notably herpes simplex virus (HSV), varicella-zoster virus (VZV), and human papillomavirus (HPV) [47]. LCs also have a sentinel role in maintaining tolerance to the bacterial skin flora and showed a protective effect against superficial fungal skin infections [47]. The decrease of LCs in patients with LMNSCI may have several clinical implications related to skin immunosuppression. The sympathetic nervous system can regulate immune responses and thus an altered activity of this system, which is common in patients with SCI, and may be responsible for the high incidence of infection in these patients. Alterations of the autonomic nervous system have been demonstrated to be to be responsible for low NK cell activity, functional impairment of T cells, and alteration of phagocytic capacity in macrophages [47]. Low expression of cellular adhesion molecules on leukocytes and reduction in both IL-2 and IL-l signaling have been observed after denervation as a result of the immune system suppression [48]. Together with these previous data, the results of our study showing a reduction in LCs content are consistent with the finding of an increased frequency of infection in these patients.

Finally, in our study we demonstrated that the functional electrical stimulation was not able to modify the content of Langerhans cells in the epidermis of the UMNSCI patients in order to restore skin immunocompetence. On the other hand, this finding is consistent with the idea that the application of an electrical rehabilitation therapy for the muscles may not be responsible for skin inflammation.

## Figures and Tables

**Figure 1 diagnostics-12-00567-f001:**
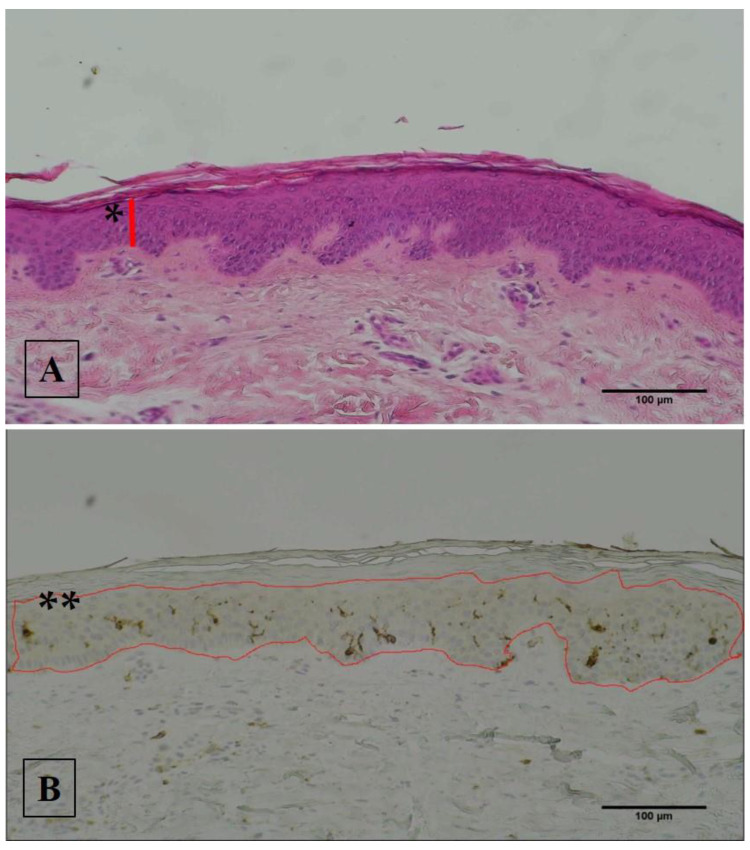
Histological sections of 5 µm thickness were collected and stained by (**A**) standard Hematoxilin-Eosin (H-E) stain protocol and (**B**) with a standard immunohistochemical procedure to analyze CD1a, a specific marker of Langerhans Cells (LCs); * = the distance between the outermost surface of the epidermis (excluding the *stratum corneum*) and the dermo–epidermal junction; ** = outline of total area of epidermis.

**Figure 2 diagnostics-12-00567-f002:**
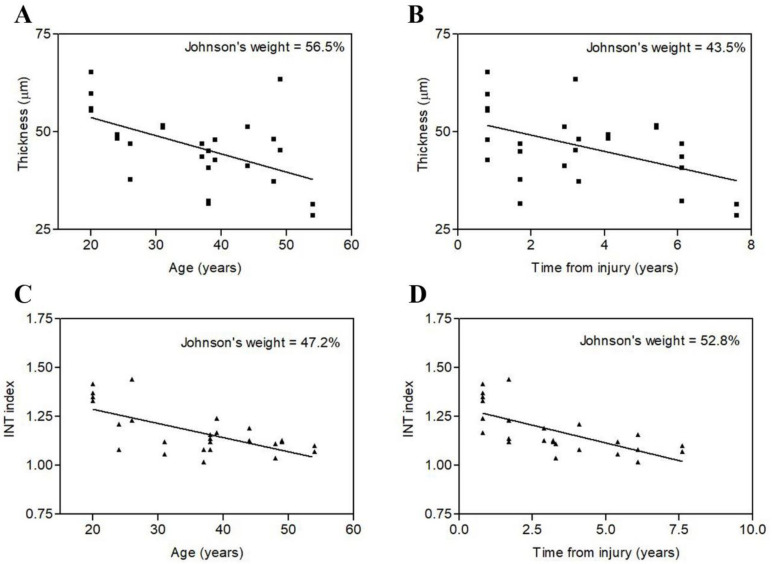
Multiple regression analysis on LMNSCI before the h-bFES, indicating a moderate (R < 0.7) but significant (*p* < 0.01, see Table 2) change of epidermis thickness with both age (**A**) and time from injury (**B**) and change of Interdigitation index (INT index) with both age (**C**) and time from injury (**D**). The indicated Johnson’s weight estimates the relative impact of each of the two factors on the multiple correlation factors.

**Figure 3 diagnostics-12-00567-f003:**
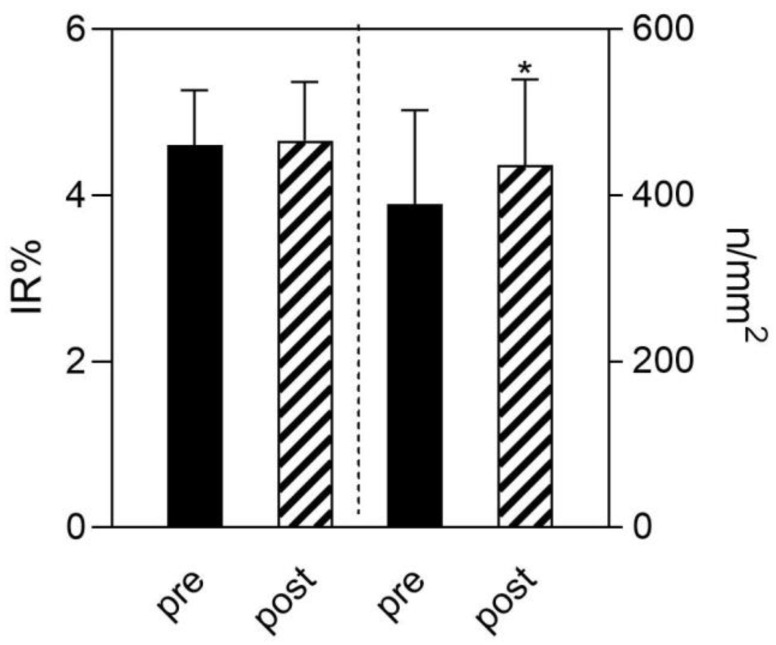
Quantitative analysis of CD1a-immunostained epidermal sections from UMNSCI patients before (pre) and after (post) 2 years of NMES. IR% represents the immunoreactive area occupied in the epidermis by Langerhans cells, while n/mm^2^ is their numerical density. Data are mean ± SEM; * = *p* < 0.05, paired *t*-test.

**Figure 4 diagnostics-12-00567-f004:**
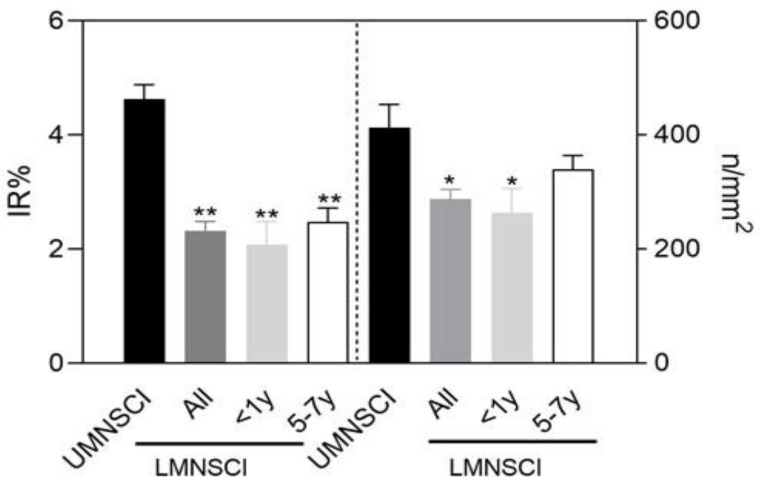
Langerhans cells (CD1a positive) in UMNSCI vs LMNSCI: Quantitation in immunostained epidermal sections of CD1a before + after (Pre + Post) 2 years of NMES on UMNSCI and before (pre) in LMNSCI. IR% is the percentage of immunoreactivity for CD1a; n/mm^2^ is the number of LC bodies per square millimeter of epidermis; data are mean ± SEM; statistical significance at *p* < 0.05 (*) *p* < 0.0001 (**), unpaired *t* test.

**Table 1 diagnostics-12-00567-t001:** Characteristics of UMNSCI patients.

Analysis	Gender		Age	Years from UMNSCI
UMNSCI.1	M	Pre NMES	20	1.8
		Post NMES	22	3.8
UMNSCI.2	F	Pre NMES	26	3.8
		Post NMES	28	3.8
UMNSCI.3	M	Pre NMES	40	3.8
		Post NMES	42	3.8

Characteristics of UMNSCI patients: 1, UMNSCI.2 and UMNSCI.3; M = male, F = female; PreNMES = before to start the NMES; PostNMES = after 2 years of NMES; age = the age before starting the NMES and after 2 years of NMES; years from UMNSCI = period from injury.

**Table 2 diagnostics-12-00567-t002:** Multiple regression analysis of the epidermis parameters (thickness and INT index) at baseline (pre-treatment) with both age and time from injury on LMNSCI before the h-bFES.

	Age		Injury Time			
	Pearson’s R	*p*	Pearson’s R	*p*	Multiple R	*p*
Epidermis thickness	−0.532	0.003	−0.487	0.006	0.595	0.007
INT index	−0.667	0.001	−0.693	0.001	0.792	0.001

**Table 3 diagnostics-12-00567-t003:** Epidermis thickness (µm): UMNSCI pre and post NMES.

Analysis	Parameter		Pre		Post	Δ%	
H-E UMNSCI.1		L	58.48		69.52		
		R	61		66.42		
	Average		59.74		67.97	12	
H-E UMNSCI.2		L	45.17		56.44		
		R	35.68		40.94		
	Average		40.43		48.69	16	
H-E UMNSCI.3		L	57.36		36.73		
		R	80.63		60.85		
	Average		69		48.79	41	
Analysis		n	pre ± SEM	n	post ± SEM	Δ%	*p*
H-E UMNSCI	thickness	6	56.39 ± 6.24	6	55.15 ± 5.50	−2.2	0.85

Morphometric analyses of epidermis: Evaluation of thickness on epidermal sections between before and after (Pre and Post) 2 years of NMES in UMNSCI. Δ% is the differential percentage of the thickness; Data are mean ± SEM; n = number of biopsies for the analysis to calculate the average. Statistical significance at *p* < 0.05, paired *t* test. Data (µm) were taken from roughly three subjects (from left L and right R leg and each average).

**Table 4 diagnostics-12-00567-t004:** Epidermis thickness (µm) in UMNSCI vs. LMNSCI.

**Epidermis Thickness (µm)—All Analyses to Date**						
UMNSCI	(Pre + Post NMES)		LMNSCI (Pre h-bFES)			
	mean ± SEM	n	mean ± SEM	n	Δ%	*p*
thickness	55.8 ± 3.97	12	46.25 ± 1.87	26	−17.1	0.018
**Epidermis thickness (µm)—UMNSCI vs. <1-year LMNSCI**						
UMNSCI	(Pre + Post NMES)		<1-year LMNSCI (Pre h-bFES)			
	mean ± SEM	n	mean ± SEM	n	Δ%	*p*
thickness	55.8 ± 3.97	12	54.66 ± 3.30	6	2.04	0.86
**Epidermis thickness (µm)—UMNSCI vs. 5 to 7-year LNMSCI**						
UMNSCI	(Pre + Post NMES)		5–7 year LMNSCI (Pre h-bFES)			
	mean ± SEM	n	mean ± SEM	n	Δ%	*p*
thickness	55.8 ± 3.97	12	40.94 ± 3.20	8	46.2	0.015

Morphometric analyses of epidermis: Evaluation of thickness on epidermal sections of pre + post 2 years of NMES in UMNSCI and pre h-bFES in LMNSCI and different periods of denervation (<1 year or 5–7 years). Δ% is the differential percentage of the thickness; data are mean ± SEM; n = number of biopsies for each analysis. Statistical significance at *p* < 0.05, unpaired *t* test.

**Table 5 diagnostics-12-00567-t005:** Epidermis Interdigitation index: UMNSCI pre and post NMES.

Analysis	Parameter		Pre		Post	Δ%	
H-E UMNSCI.1		L	1.57		1.58		
		R	1.38		1.38		
	Average		1.47		1.48	0.39	
H-E UMNSCI.2		L	1.19		1.19		
		R	1.53		1.45		
	Average		1.36		1.32	−2.91	
H-E UMNSCI.3		L	1.44		1.13		
		R	1.43		1.61		
	Average		1.43		1.37	−4.13	
Analysis		n	pre ± SEM	n	post ± SEM	Δ%	*p*
H-E UMNSCI Int. index		6	1.42 ± 0.05	6	55.15 ± 5.50	−2.2	0.63

Morphometric analyses of epidermis: Evaluation of interdigitation index on epidermal sections between before and after (Pre and Post) 2 years of NMES in UMNSCI. Δ% is the differential percentage of the index; data are mean ± SEM; n = number of biopsies for the analysis. Statistical significance at *p* < 0.05, paired *t* test. Data (µm) are taken from three subjects (from left L and right R leg and average).

**Table 6 diagnostics-12-00567-t006:** Epidermis Interdigitation index in UMNSCI vs. LMNSCI.

**Epidermis Interdigitation Index—All Analyses to Date**						
UMNSCI	(Pre + Post NMES)		LMNSCI (Pre h-bFES)			
	mean ± SEM	n	mean ± SEM	n	Δ%	*p*
INTindex	1.41 ± 0.05	12	1.17 ± 0.02	26	−20.2	<0.0001
**Epidermis Interdigitation index—UMNSCI vs. <1-year LMNSCI**						
UMNSCI	(Pre + Post NMES)		<1-year LMNSCI (Pre h-bFES)			
	mean ± SEM	n	mean ± SEM	n	Δ%	*p*
INTindex	1.41 ± 0.05	12	1.31 ± 0.037	6	7.6	0.20
**Epidermis Interdigitation index—UMNSCI vs. 5 to 7-year LNMSCI**						
UMNSCI	(Pre + Post NMES)		5–7 year LMNSCI (Pre h-bFES)			
	mean ± SEM	n	mean ± SEM	n	Δ%	*p*
INTindex	1.41 ± 0.05	12	1.09 ± 0.014	8	−29.8	<0.0001

Morphometric analyses of epidermis interdigitation index: Evaluation of index on epidermal sections before + after (Pre + Post) 2 years of NMES in UMNSCI and before (pre) in LMNSCI at different periods of denervation (<1 year and 5–7 year). Δ% is the differential percentage of the index; data are mean ± SEM; n = number of biopsies for each analysis. Statistical significance at *p* < 0.05, unpaired *t* test.

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
