# Peer review of "Trauma of Peripheral Innervation Impairs Content of Epidermal Langerhans Cells"

_diagnostics, 2022, doi:10.3390/diagnostics12030567_

Round 1

Reviewer 1 Report

Dear Authors,

Attempts to observe patients for a long time look good. However, as this is a human experiment, data are limited and information is limited. If this paper is accepted as a case report, it would be meaningful, but it is unclear how much of a ripple effect it will have. And, swelling of the patched area is natural with or without electrical stimulation. Langerhans cells were followed, but the meaning is also unclear. I think more experiments are needed. 

It is recommended to add an in-vivo experiment. Even if it is not for a long time, I think that if the experimental plan is well prepared, you will be able to see the difference. 

Best,

Author Response

Dear Authors,

Attempts to observe patients for a long time look good. However, as this is a human experiment, data are limited and information is limited. If this paper is accepted as a case report, it would be meaningful, but it is unclear how much of a ripple effect it will have. And, swelling of the patched area is natural with or without electrical stimulation. Langerhans cells were followed, but the meaning is also unclear. I think more experiments are needed. 

It is recommended to add an in-vivo experiment. Even if it is not for a long time, I think that if the experimental plan is well prepared, you will be able to see the difference. 

    We agree with the referee that in vivo studies on a larger number of patients are necessary and this has been underlined in the conclusions of the article.

Reviewer 2 Report

The authors studied the influence of deinnervation and subsequent artificial electric stimulation on epidermal thickness. The study is of some relevance, however, I suggest being more focused on the major findings. E.g. the abstract and introduction should be shortened. Also, the results could be demonstrated in a more condensed way. I missed the histological pictures to document the major findings - a general histology overview is not enough. 

The authors have also to be more humble while describing their results. Such words, like "these ore original relevant results", "we definetely proved". It is the right of the readers to judge the relevance of the results. 

The abbreviation should be explained while used for the first time, both in the abstract, as well as in the main text.

Author Response

The authors studied the influence of deinnervation and subsequent artificial electric stimulation on epidermal thickness. The study is of some relevance, however, I suggest being more focused on the major findings. E.g. the abstract and introduction should be shortened. Also, the results could be demonstrated in a more condensed way. I missed the histological pictures to document the major findings - a general histology overview is not enough. 

The authors have also to be more humble while describing their results. Such words, like "these ore original relevant results", "we definetely proved". It is the right of the readers to judge the relevance of the results.

The abbreviation should be explained while used for the first time, both in the abstract, as well as in the main text.

We have shortened both the introduction section and the abstract. Furthermore the results have been described in a more humble and condensed way (please see Abstract, Introduction and Result sections). Abbreviations have been explained when used for the first time.

Reviewer 3 Report

The paper approaches an important topic, however in a confuse text and form.

The introduction is confused in the topic and the aims are not clear.

In the methodology section, I did not understand the reason why authors must use different statistical software. Authors are invited to revise or explain this option. Another point that should be revised is the text of the lines 111 and 112 and Table 1. Did authors want to say: Biometric physical characteristics?

It is not clear (in the text and the Table 1) the groups division into UMNSCI.1, 2 and 3. Authors are invited to clarify and improve legend.

In the figure 1. Is not clear the CD1a staining and the specific marker of LCs.

Is not clear, also the separation between methodology and results.

Table 2. - Concerning the Pearson’s correlation is difficult to accept significance and relevant correlation below 0.7-0.8. The correlation values must have the attention and justification of authors.

Authors are invited to revise the statistic of the Table 3. It is very strange that authors obtained a p <0.0001 since results indicated another tendency. Also Figure 3 have statistic problems. The last column is significant?

I cannot analyze the remaining tables since they are unformatted.

Instead of referring several times to previous works, authors are invited to show some of the obtained results since they are relevant to the present study.

The authors are invited to revise the discussion and conclusions. Conclusions must agree with the obtained results.

Author Response

The paper approaches an important topic, however in a confuse text and form.

The introduction is confused in the topic and the aims are not clear.

    The introduction section has been shortened and modified.

In the methodology section, I did not understand the reason why authors must use different statistical software. Authors are invited to revise or explain this option.

    Statistical analyses were performed using GraphPad Prism 5.0 software and this has been more clearly explained in the Material and Methods section

Another point that should be revised is the text of the lines 111 and 112 and Table 1. Did authors want to say: Biometric physical characteristics?

    The word “biometric” has been deleted from the manuscript

It is not clear (in the text and the Table 1) the groups division into UMNSCI.1, 2 and 3. Authors are invited to clarify and improve legend.

    Group division has been clarified and the legend of Table 1 modified accordingly.

In the figure 1. Is not clear the CD1a staining and the specific marker of LCs.

    The specific marker of LCs is the CD1a molecule and this has been explained in the legend to figure 1

Is not clear, also the separation between methodology and results.

    The separation between these two sections has been better indicated

Table 2. - Concerning the Pearson’s correlation is difficult to accept significance and relevant correlation below 0.7-0.8. The correlation values must have the attention and justification of authors.

Authors are invited to revise the statistic of the Table 3.

It is very strange that authors obtained a p <0.0001 since results indicated another tendency.

Also Figure 3 have statistic problems. The last column is significant?

    Tables and Figures have been modified in order to show in a more comprehensible the results of the study including the statistical analysis. In order to avoid confusion, the old Table III which contained the results of a previous investigation has been eliminated.

I cannot analyze the remaining tables since they are unformatted.

    The relevant Tables have been formatted

Instead of referring several times to previous works, authors are invited to show some of the obtained results since they are relevant to the present study.

The authors are invited to revise the discussion and conclusions. Conclusions must agree with the obtained results.

    Discussion and conclusions have been revised

Round 2

Reviewer 2 Report

None

Author Response

The referee did not provide comments and/or suggestions

Reviewer 3 Report

The version now presented has substantially improved the quality of the work. However, the Introduction section can and should be improved. The discussion uses concepts that could have been in the introduction.

The last two paragraphs of the introduction should be removed as they are out of context. Instead of these concepts, the objectives of the work should be clearly stated.

Methodology and results are now more comprehensive.

Nevertheless, the discussion does not address the most relevant results obtained by the authors. Therefore, I invite the authors to review the discussion. Moreover, the conclusions must be more explicit since do not reflect the obtained results.

Minor points

Pag.9 “to be” is repeated.

Author Response

1) The last two paragraphs of the introduction should be removed as they are out of context. Instead of these concepts, the objectives of the work should be clearly stated.

  • The last two paraghraps of the introduction have been now removed and the objectives of the work have been more clearly stated (please see page 2, lines 20-23)

2) Nevertheless, the discussion does not address the most relevant results obtained by the authors. Therefore, I invite the authors to review the discussion. 

The discussion has been now modified in order to underline the most relevant results (please see page 9, lines 4-6 and page 10 lines 2-5). Furthermore, the last paragraph has now been removed.

Round 3

Reviewer 3 Report

Thanks to authors to considering our suggestions.

The manuscript improved significantly since the first version.

The authors are invited to verify spelling.

This manuscript is a resubmission of an earlier submission. The following is a list of the peer review reports and author responses from that submission.

Round 1

Reviewer 1 Report

Albertin et al. study entitled "Long-term complete lower motor neuron traumatic spinal cord injury impairs trophism and Langerhans cells of the epidermis more than upper motor neuron traumatic lesion." is a very interesting piece of work in the field of SCI and skin de-innervation.  As manuscript work is interesting but it lacks proper writing and data presentation. To improve the presentation I have some suggestions and some questions that need to be address 1. There are no sections needed in the introduction and Langerhans cells' role should be introduced at the beginning of the manuscript, i.e. from abstract to introduction. 2. Write proper keywords.  3. Restructure the introduction as there is no need for materials and methods there.   Instead, introduce the hypothesis at the end of the introduction.  4. Properly write the materials and methods without mentioning any results one by one. 5. Results need to be clearly defined without mixing or duplicating the methods. No need for figure one. Instead, present some important data from the tables as a figure where you can show comparative images of all the groups and bar diagrams of quantitative data especially the comparison between UMN and LMN T-SCI with or without stimulation. This should include the HE and CD1a staining images. Some tables needed proper data arrangement.  6. Discussion is very week and needs re-writing.  Some data from animal studies should be included for improving the discussion. Other cells are playing a role in skin atrophy and other related factor. A few studies have studied the role of upper and lower skin changes after SCI in animals, including pressure ulcers healing. Authors should consider adding that information to the discussion.   

Reviewer 2 Report

Using an iThenticate software, I checked plagiarism in the manuscript. The manuscript contains the same sentences and phrases as internet sources, especially in METHODS. I guess it is a kind of "self-plagiarism". So I will not go ahead.